# The Natural Oligoribonucleotides Functionalized by D-Mannitol Affected Interactions of Hemagglutinin with Glycan Receptor Indicating Anti-Influenza Activity

**DOI:** 10.3390/membranes11100757

**Published:** 2021-09-30

**Authors:** Zenoviy Tkachuk, Nataliia Melnichuk, Roman O. Nikolaiev, Kosma Szutkowski, Igor Zhukov

**Affiliations:** 1Institute of Molecular Biology and Genetics, National Academy of Sciences of Ukraine, 03680 Kyiv, Ukraine; natalia.melnichuk8@gmail.com (N.M.); romanfromukrain@gmail.com (R.O.N.); 2“Biocell” Subsidiary Company of Biocell Laboratories Inc. (USA), Klovskiy Uzviz 17, 03680 Kyiv, Ukraine; 3NanoBioMediacal Centre, Adam Mickiewicz University, Ul. Wszechnicy Piastowskiej 3, 61-614 Poznan, Poland; kosma.szutkowski@outlook.com; 4Institute of Biochemistry and Biophysics, Polish Academy of Sciences, Ul. Pawińskiego 5a, 02-106 Warsaw, Poland

**Keywords:** hemagglutinin, oligoribonucleotides, mannitol, molecular docking

## Abstract

Hemagglutinin (HA), the class I influenza A virus protein is responsible for the attachment of virus particles to the cell by binding to glycan receptors, subsequent virion internalization, and cell entry. Consequently, the importance of HA makes it a primary target for the development of anti-influenza drugs. The natural oligoribonucleotides (ORNs) as well as their derivatives functionalized with D-mannitol (ORNs-D-M) possess anti-influenza properties in vitro and in vivo due to interaction with HA receptor sites. This activity suppresses the viral infection in host cells. In the present work, the complexes of ORNs and ORNs-D-M with HA protein were studied by agglutination assay, fluorescence spectroscopy, as well as molecular docking simulations. Acquired experimental data exhibited a decrease in HA titer by 32 times after incubation with the ORNs-D-M for 0.5–24 h. Quenching fluorescence intensity of the HA suggests that titration by ORNs and ORNs-D-M probably leads to changes in the HA structure. Detailed structural data were obtained with the molecular docking simulations performed for ORNs and ORNs-D-M ligands containing three and six oligoribonucleotides. The results reveal that a majority of the ORNs and ORNs-D-M bind in a non-specific way to the receptor-binding domain of the HA protein. The ligand’s affinity to the hemagglutinin was estimated at the micromolar level. Presented experimental data confirmed that both natural ORNs and functionalized ORNs-D-M inhibit the interactions between HA and glycan receptors and demonstrate anti-influenza activity.

## 1. Introduction

Influenza epidemic activity causes significant morbidity and mortality. It increases the costs of health services and contributes to economic losses due to absence from work. The World Health Organization (WHO) estimated that the 2009 pandemic alone caused 100,000–400,000 deaths, not only among groups considered to be at a higher risk of complications, such as the elderly, persons with chronic conditions, and pregnant women but also in the group of young, healthy individuals. The annual trivalent or quadrivalent vaccines are the most effective way to prevent infection and severe outcomes caused by influenza viruses. However, due to rapid antigenic drift and shift in influenza viruses, the selection of appropriate vaccine strains is a formidable task [1,2,3]. In connection with the emergence of resistance to existing licensed anti-influenza drugs and the constant threat of an epidemic or pandemic of influenza virus, it is an urgent need to search for a new antiviral drug against new targets [4,5,6].

Influenza virion contains eight proteins which are surrounded by a protein coat that includes two major surface glycoproteins—hemagglutinin (HA) and neuraminidase. HA, the glycoprotein is located on the surface membrane of the influenza virus and is an integral part of its infectivity [7]. HA coding as the precursor (HA_0_) polypeptide which after cleavage by cellular proteases, generates the head (HA_1_) and stem (HA_2_) subunits linked by a single disulfide bond. The 3D structure of the HA presents as a symmetric trimer, with each monomer comprising a receptor-binding domain (RBD) interacting with sialylated glycan receptor (HA_1_), together with membrane fusion subdomains (HA_2_) [8,9] (Figure 1). The N-terminal fragment of HA_2_ subdomain constitutes the influenza fusion peptide inserted into the endosomal membrane during virus entry [9]. HA has a multifunctional activity and plays a critical role in viral binding, entry, and fusion processes into the host cell [10], therefore, it is a promising target for developing anti-influenza drugs acting as the initial entry blockers of the viral life cycle. However, the changeable nature of the HA creates difficulties in drug development [11]. Therefore, it is important to develop an anti-influenza drug based on non-specific interactions with hemagglutinin.

Natural oligoribonucleotides (ORNs) presented in total yeast RNA, together with their functionalized with D-mannitol (ORNs-D-M) variant, demonstrated antiviral activity against a wide range of DNA, and RNA viruses [12,13]. As reported previously, ORNs and ORNs-D-M show anti-influenza activity in vitro and in vivo [14] inhibiting the influenza-induced expression of innate immunity genes leading to reduce the influenza-mediated immunopathology [15]. The mechanism of inhibition infectivity of the influenza virus by ORNs-D-M based on blocking HA–glycan interactions has been proposed [16,17].

In this report, we demonstrate anti-influenza action of the ORNs and ORNs-D-M based on non-specific binding with moderate (micromolar) affinity to the hemagglutinin. The docking simulations performed by AutoDock Vina procedure [18] reveals that the binding site for the ORNs ligands is placed at the RBD domain, which is responsible for the interaction with glycan receptor. The ORNs and ORNs-D-M binding probably cause tiny changes in the spatial structure of HA. However, the detailed mechanism of the interaction remains unknown and requires further research.

## 2. Materials and Methods

### 2.1. Materials

Oligoribonucleotides (ORNs)—total yeast RNA with the dominant fraction of 3–8 nucleotides. ORNs-D-Mannitol (ORNs-D-M) complexes—total yeast RNA is modified with D-mannitol (D-M) in a mixing ratio of 2.5:1.0 mg. The RNA drugs (ORNs and ORNs-D-M complexes) were purchased from Goodwill Associates, Washington, DC, USA. Vaxigrip vaccine (Sanofi Pasteur, France) and used to study the interaction of the RNA drugs with the HA of the influenza virus. “Vaxigrip” is a split influenza virus, namely the surface antigen of HA at a concentration of 45 μg/0.5 mL (influenza viruses A/Michigan/45/2015) (H1N1) pdm09-like (A/Michigan/45/2015, NYMC X-275), A/Hong Kong/4801/2014 (H3N2)-like, (A/Hong Kong/4801/2014, NYMC X-263B), and B/Brisbane/60/2008-like (B/Brisbane/60/2008, wild type), which were cultured on chicken embryos of healthy chickens).

### 2.2. Hemagglutination Analysis

To evaluate the ability of the ORNs and ORNs-D-M complexes to inhibit the interaction between the HAs and glycans, we conducted the hemagglutination assay. The assays were performed on a round-bottomed 96-well plate using 1% of human erythrocytes 0 (I) in phosphate-buffered saline (PBS). Fifty microliters of PBS buffer (pH 7.4) were added to each well. The HA (225 ng/0.1 mL) was incubated with the ORNs (2.5 mg/mL), D-M (1 mg/mL) (SIGMA, St. Louis, MO, USA), the ORNs-D-M (0.8, 1.7, 3.5 mg/mL) at 20 °C for 30 min. In the first well of column, 50 μL of control or test sample were added. Each good sample was mixed and 50 μL were transferred to the next well on its right. Mixing was repeated and 50 μL were transferred down the length of the plate. The 50 μL from the last well was discarded. To each well, 50 μL of 1% erythrocytes working solution were added and mixed gently. The reaction mixtures were incubated at room temperature for 60 min to allow hemagglutination, followed by photography to document the results. HA control was incubated without drugs at 20 °C for 30 min. The negative results appeared as dots in the center of round-bottomed plates. The positive results formed a uniformly reddish color across the well. The HA titer was determined as the number of the highest dilution factor that produced a positive reading. To estimate the duration of the HA–ORN-D-M interaction, the HA (225 ng/0.1 mL) was incubated with the ORNs-D-M (3.5 mg/mL) for 30 min, 1, 2, 4, and 24 h at room temperature, followed performed the hemagglutination assay.

### 2.3. Fluorescence Measurements

The amino acids tryptophan (Trp), tyrosine (Tyr), and phenylalanine (Phe) are fluorescent in nature, thus contributing to the fluorescence of protein in which they are present [19]. The fluorescence spectrum of the HA fluorescence intensity quenching by the ORNs and ORNs-D-M was measured using JASCO FP-8200 spectrofluorometer (Jasco, Tokyo, Japan). The 3D and fluorescence spectra of the HA, ORNs, ORNs-D-M, HA-ORN, and HA-ORN-D-M were recorded at room temperature. The studies were performed in PBS buffer (Sigma Aldrich, St. Louis, MO USA), pH 7.4. The 3D fluorescence spectra (total fluorescence of the sample) of the HA protein and the ORNs and ORNs-D-M ligands were measured to determine the maximum of their excitation and the possibility of overlapping spectra. Fluorescence measurement of total HA (0.15 μM) and the ORNs, ORNs-D-M (54.3 μM) was performed at the following conditions: excitation wavelength—$\lambda_{Ex}$ of 200–400 nm; measurement wavelength—$\lambda_{Em}$ 210–750; slit width—5 nm; scanning speed—10,000 nm/min; detector voltage—500 V.

Fluorescence spectra of the HA during titration with the ORNs and ORNs-D-M were recorded for the following parameters: $\lambda_{Ex}$ of 275 nm; $\lambda_{Em}$ of 280–450 nm; data interval—1 nm; slit width—2.5 nm; scanning speed—100 nm/min; the voltage of the detector—500 V. The HA at a concentration of 0.15 μM was titrated with 5 μL of 1 mM ORNs and ORNs-D-M until complete quenching of the fluorescence signal. After that, the 3D fluorescence spectra were recorded, which aided in observe any changes in the HA structure due to interaction with the ORNs and ORNs-D-M.

Construction spectra of the HA fluorescence quenching by the ORNs and ORNs-D-M and calculations of the dissociation constant ($K_{d}$) were performed using the program Origin 8.1 (OriginLab, USA) by the following expression:

$$K_{d}=\frac{\left(1-\theta \right)(D-\left(\theta \cdot P_{0}\right)}{\theta }$$
where θ is the fraction of bound to total protein at the stoichiometric point and $P_{0}$ the total protein concentration in the cuvette. *D*—ligand concentration at any titration point [20].

### 2.4. Docking of the ORNs and ORNs-D-M to Influenza Hemagglutinin

The docking procedure was performed with AutoDock Vina program [18] included in Yasara software (version 20.8.23) [21]. As a starting point, the 3D structure of the H3 strain of the HA protein was extracted from the A/Hong Kong/1/1968 (H3N2) influenza virus, solved with 2.6 Åresolution by X-ray diffraction (pdb 5t6n) (Figure 1) [22]. Taking into account a nucleotide length distribution detected for ORNs and ORNs-D-M, there are four ligands that were docked to the HA3—the ORNs with three (A3-ORNs) and six (A6-ORNs) units. The same procedure was applied for the variants functionalized with D-mannitol–A3-ORNs-D-M, and A6-ORNs-D-M (Appendix A). There are five runs of docking algorithm “VINA” with AMBER03 forcefield with 32 substrates in each round were performed at temperature 298 K. The simulations were conducted by the macro “dock_runensamble”, included in the Yasara Structure pack.

## 3. Results and Discussion

### 3.1. Analysis of the Interaction of ORNs and ORNs-D-M with HA

To study the interaction of ORNs and ORNs-D-M with the HA influenza virus, we used the isolated hemagglutinin in the form of vaccine “Vaxigrip”. Based on this hemagglutination analysis the HA titers were calculated as the reciprocal of the minimum dilution forming the red button, indicating HA unit (HAU). It was shown that the HA titer of the isolated HA at a concentration of 225 ng/0.1 mL was 65 HAU/0.1 mL (Figure 2). After incubation of the HA with the ORNs (2.5 mg/mL) and the ORNs-D-M (3.5 mg/mL) for 30 min at room temperature, the HA titer decreased 32 times compared to the HA control (*p* < 0.05). We have also investigated the effect of ORNs-D-M on the HA activity in the series of concentrations. The ORNs-D-M complexes at a concentration of 1.7 mg mL reduced the HA activity by 32 times, while the erythrocytes agglutination by the HA, pre-incubated with ORNs-D-M at a concentration of 0.8 mg/mL, remained unchanged in comparison to the HA control. The ORNs, ORNs-D-M, and D-M have been studied not to affect the erythrocytes during agglutination [16].

Subsequently, we estimated the minimum concentration of HA—3.5 ng/0.1 mL—at which the protein can agglutinate erythrocytes at a concentration of 1% (Figure 2). The ORNs-D-M complexes at concentrations of 1.7 and 3.5 mg/mL interact with ∼56 ng/0.1 mL of the HA.

The hemagglutination assay demonstrates the presence of the long-term interaction ORNs-D-M with the HA protein exhibiting a decrease HA activity by 32 times after pre-incubation for 30 min, 1, 2, 4, and 24 h compared to the control (Figure 3).

### 3.2. HA Fluorescence Quenching by the ORNs and ORNs-D-M

Fluorescence spectroscopy was used to measure the effect of ORNs and ORNs-D-M on the HA structure. Total fluorescence spectra of the HA, ORNs, and ORNs-D-M showed that the protein and ligands do not overlap with “landscape” spectra, indicating that the electron reabsorption effect can not occur during protein–ligand interaction study.

The 3D spectrum of HA fluorescence is found out to have two emission peaks. The first emission peak (I) with a maximum excitation $\lambda_{Ex}$ 275 nm corresponds to the HA base (characterizes Trp and Tyr residues in HA), while the second peak (II)—with a maximum excitation $\lambda_{Ex}$ 225 nm—corresponds to aromatic residues that have reached above the excited electronic states [23] (Figure 4). Any changes in the intensity and positioning of the peak I strongly suggest the presence of an alteration on the microenvironment of tryptophan and tyrosine residues [24].

An additional fluorescent component (peak III on Figure 4B,C), appeared near peak I under saturation by ORNs and ORNs-D-M, suggesting that ligand binding leads to tiny structural changes of the HA protein. At the same time, 3D fluorescence spectra of the HA–ORNs and HA–ORNs-D-M complexes showed effect fluorescence quenching for both HA peaks, which do not change their location in the three-dimensional coordinate system (Appendix A) [24]. Nevertheless, collected experimental data demonstrated that the effect of quenching is more significant for the functionalized (ORNs-D-M) version of the ligand (Figure 5).

The dissociation constants for the ORNs and ORNs-D-M ligands were evaluated on base experimental data (Appendix A). The $K_{d}$ around $10^{-7}$–$10^{-6}$ M were obtained for both substrates, which can be defined as a moderate level of dissociation.

### 3.3. ORNs and ORNs-D-M Bind to the HA Protein Mostly in RBD Subdomain

To explore details of binding oligoribonucleotides to hemagglutinin we performed the docking procedure of the ORNs and ORNs-D-M to the H3 strain of the HA trimer (HA3). It was solved by X-ray crystallography with resolution 2.6 Åhigh-resolution 3D structure of the HA3 from pandemic H3N2 (A/Hong Kong/1968) influenza virus (pdb 5t6n) [22].

The ORNs and ORNs-D-M substrates used in our study exhibit a broad distribution in the length of nucleotides chains. According to our preliminary data, the distribution of oligoribonucleotides reveals variation from 1 up to 20 nucleotides or even longer. Nevertheless, the majority of the ORNs comprise between three and nine units with maximum populated chains with six nucleotides. THe length of the ORNs and functionalized ORNs-D-M can be an additional parameter regulated process of binding substrates to the HA protein, which we explored by the docking simulations with AutoDock Vina algorithm [18]. Taking into account distribution of chains length there are two types of ligands, with three (A3-ORNs) and six (A6-ORNs) nucleotides, together with their functionalized by D-mannitol (A3-ORNs-D-M and A6-ORNs-D-M) variants were selected as a reasonable choice for representation of whole ORNs and ORNs-D-M ensemble (Appendix A). Each run of simulations includes docking of the 32 substrates to the HA3 hemagglutinin. Visualized results demonstrated that all ORNs and functionalized by D-mannitol ORNs-D-M ligands occupy the various fragments of the HA3 structure (Appendix A). In spite of the majority bound oligoribonucleotides observed in the RBD fragment of the HA_1_ subunit, some complexes contain ORNs or ORNs-D-M in the stem region (HA_2_ subunit). Contrary to the HA_1_, the HA_2_ undergoes from a globular fold to three extended loops with central coil–coil structure under transition between prefusion to postfusion conformation [25].

Analysis of docking results was performed on four complexes of ligands that were selected basing on the low-energy criteria. The evaluated data showed that either ORNs or ORNs-D-M bind to the HA3 protein mostly at the membrane-distal end of the structure, inside the RBD fragment. The $K_{d}$ value obtained in the docking procedure is in the range of $10^{-8}$–$10^{-4}$ M suggested that interactions between ORNs and ORNs-D-M and hemagglutinin are not specific and characterized by the moderate level of affinity (Appendix A). Extracted $K_{d}$ values are in line with previously on-base experimental data recorded by fluorescence spectroscopy (Appendix A). It is interesting that a similar $K_{d}$ was reported for binding Arbidol [22] and TBHQ [10] substrates to the HA protein from H3N2 and H14N5 influenza viruses, respectively.

Our data suggest a model of the multiple binding ORNs or ORNs-D-M ligand to the one HA that can dramatically amplify the effect of the hemagglutinin’s inhibition interaction with glycan receptor. For instance, a cavity characterized by a volume nearly 3000 Å^3^ was formed by three HA_1_ subunits inside the RBD domain (Figure 1). The docking procedure confirms the existence of the HA complex with two A3-ORNs or A3-ORNs-D-M substrates where the cavity is occupied by the two ligands (Appendix A).

The 3D structures evaluated by docking procedure reveals that the majority of the ORNs and ORNs-D-M ligands in complexes with HA3 protein binding inside the RBD fragment of HA_1_ subunit. Moreover, in many cases, the position substrates are close to the epitopes which are recognized by native human anti-bodies FluA-20 [26]. Based on the structural data, conserved residues involved in the interactions with glycan receptors in the hemagglutinin revealed the interface for binding sialic acid located at the membrane-distal tips of the RBD domain. The same epitopes are recognized specifically by IgG-type [27] or monoclonal mAb6-9-1 anti-bodies [28]. For the HA3 strain the conserved motif comprised α-helix (residues ^190^EQTNLYVQA^199^) together two loops (residues ^130^VTQNGGSNACK^140^, and ^220^RPWVGQSGRI^230^) on the two sides of central helix (numbering according to the sequence of HA3) (Figure 1) [8].

Decreased activity of HA by 32 times after incubation with the ORNs and ORNs-D-M was detected by agglutination assay (Figure 3). That anti-influenza effect is achieved by inhibition HA interaction with glycan receptor due to binding about 500 ORNs or ORNs-D-M molecules per influenza virion [29]. The small structural alterations of the hemagglutinin in complexes with the ORNs and ORNs-D-M suggest a new fluorescent component (peak III) observed on 3D fluorescence spectra due to interaction with the ORNs and ORNs-D-M [24].

The moderate values of $K_{d}$ observed in fluorescence experiments, confirmed by docking simulations with AutoDock Vina protocol, suggest a non-specific character for binding substrates to the HA glycoprotein. The HA3–ORNs and HA3–ORNs-D-M complexes exhibited three preferable localizations of the ORNs and ORNs-D-M ligands on the 3D structure. The majority of substrates bind to a membrane-distal fragment of HA, occupying positions in the RBD subdomain. In most cases, oligoribonucleotides show contacts with epitopes forming a conserved motif for interaction HA with glycan receptor (Figure 6), effectively block the HA–glycan interaction [16]. Structural analysis binding interface for low-energy structures of complexes reveals the existence of tryptophan residues in distances less than 5 Å, which can explain changes observed in the fluorescence data.

Another important aspect of functioning HA glycoprotein is which affects receptor-binding specificity. The effect of post-translational glycosylation for functioning HA glycoprotein has been known for years and is well documented (Appendix A) [30,31]. The 3D structure of H3N2 A/Hong Kong/1968 (pdb 5t6n) evaluated by X-ray crystallography includes various oligosaccharides in several positions (Appendix A). Analysis sequence of the HA3 hemagglutinin with GlyProt database (http://www.glycosciences.de/modeling/glyprot/php/main.php accessed 11 May 2021) shows the 18 sites possible for glycosylation. The 15 out of 18 possible epitopes are located at the membrane-distal end of the hemagglutinin structure. It is nteresting that two glycosylation positions—Asn81 and Asn165—are close to the glycan-binding consensus epitopes which are critical for interaction with the glycan receptor (Appendix A). It is clear that binding ORNs and their functionalized version ORNs-D-M provide to downregulation of post-translational glycosylation and block HA activity by inhibiting interactions HA with glycan receptor.

The HA3 complexes with ORNs and ORNs-D-M where ligands occupied positions in the stem region (HA_2_ subunit) were also detected. The low-energy structures characterized by such position of the A3-ORNs, or A6-ORNs and A6-ORNs-D-M are presented in Figure 7A,B and Figure 8A,B. Localization substrates in that region stabilize the HA3 structure preventing transition HA_2_ subunit of the hemagglutinin. In particular, the ORNs and ORNs-D-M mechanically block the rearrangement of fusion peptides located in the N-terminal fragment of HA_2_ and effectively inhibit the process of viral fusion saving HA glycoprotein in the prefusion state (Appendix A).

Docking for the ORNs and ORNs-D-M ligands exhibited diverse localization species on the HA structure. Moreover, taking into account distribution in length oligoribonucleotide sequences and their functionalized variant we can suggest an effective anti-influenza mechanism due to the non-specific character of interactions. In fact, we conclude that influenza virion can bind a hundred ligands with various lengths and positions on HA structure. In such a case, the ORNs and ORNs-D-M substrates can block both natural activities of the HA glycoprotein–interaction with glycan-receptor and integration to the host cell. The $K_{d}$ values of nearly 10 μM and non-specific character of binding suggest that a large dose has to be provided to obtain a strong anti-influenza effect.

## 4. Conclusions

Presented experimental data demonstrate that natural oligoribonucleotides and their derivatives functionalized by D-mannitol bind in a non-specific way to the hemagglutinin with moderate affinities of around 10 μM. We conclude that ORNs and ORNs-D-M would affect the interactions between HA and glycan receptors and pose a possible pathway for anti-influenza activity. In our opinion, natural oligoribonucleotides and their functionalized derivatives constitute promising ligands for future studies as novel anti-influenza drugs. Taking into account the recently published data describing the transition between two structural forms affected by N-glycosylation of SARS-CoV-2 spike protein regulating interaction with the angiotensin-converting enzyme 2 receptor (ACE2) in specific positions, we can speculate that ORNs functionalized by D-mannitol can present activity against the COVID-19-CoV-2 pandemic virus [32].

## Figures and Tables

**Figure 1 membranes-11-00757-f001:**
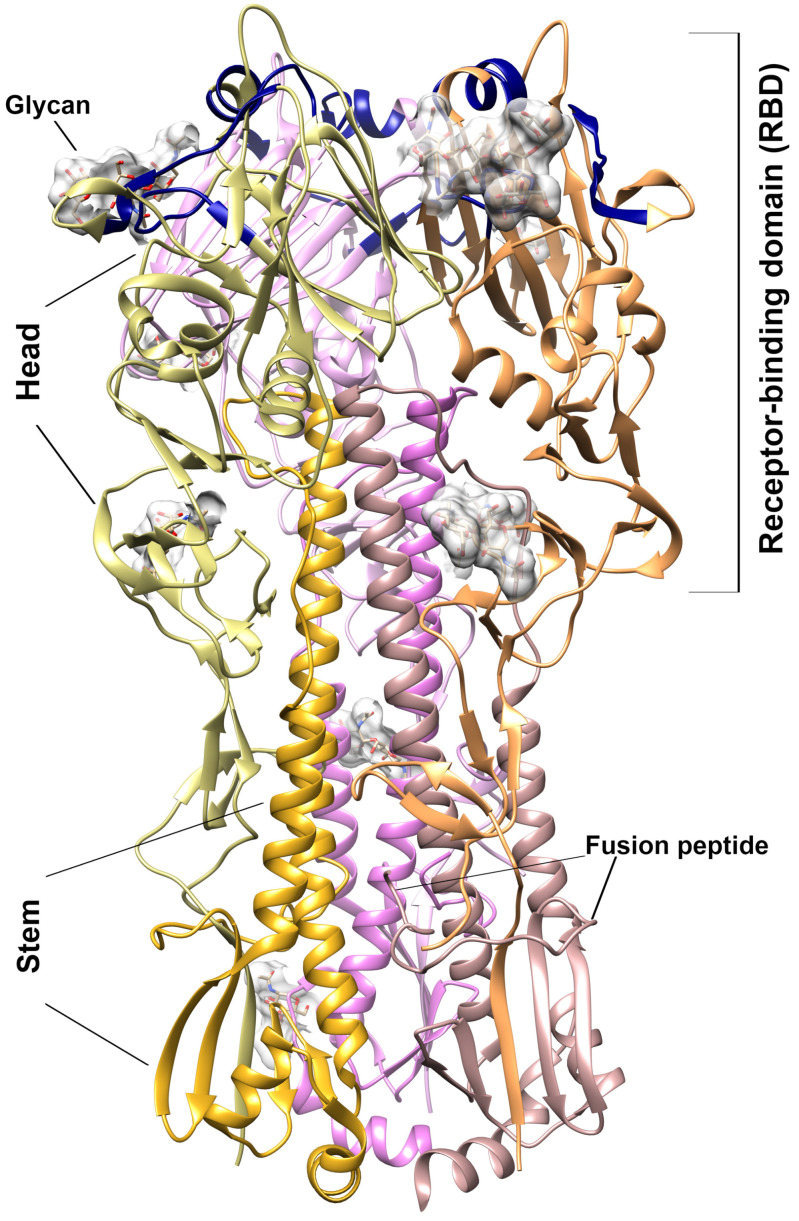
3D structure of the hemagglutinin from H3N2 A/Hong Kong/1968 (pdb 5t6n). Ribbon presentation of the trimer structure. The HA_1_ and HA_2_ subunits are discriminated by different colors. The glycan moieties presented in the solved structure are shown by the gray surface. The receptor-binding site—130 loop (epitopes 130–140), 190-α-helix (epitopes 190–199), and 220-loop (epitopes 220–230)—is highlighted by dark blue.

**Figure 2 membranes-11-00757-f002:**
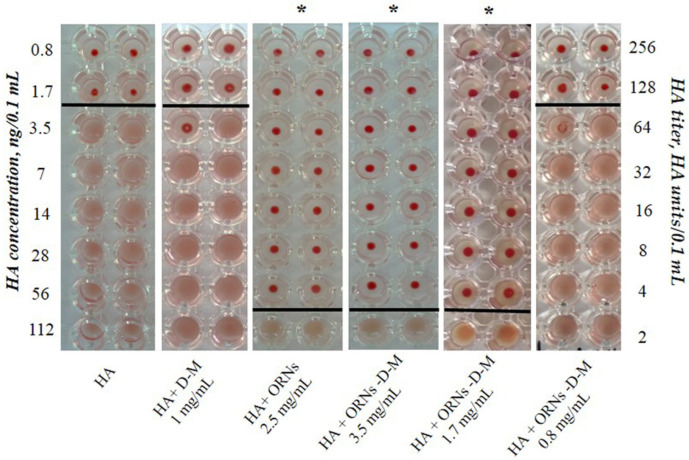
The erythrocyte agglutination mediated by the HA proteins (Vaxigrip). View of the hemagglutination reaction mixtures displayed in a round-bottomed microwell plate after the addition and dilution of the HA, which was pre-incubated with ORNs and ORNs-D-M in concentrations of 0.8–3.5 mg/mL. The picture demonstrates one of five independent experiments. Statistical significance was evaluated using the Student’s *t*-test, relative to the control HA (*p* < 0.05).

**Figure 3 membranes-11-00757-f003:**
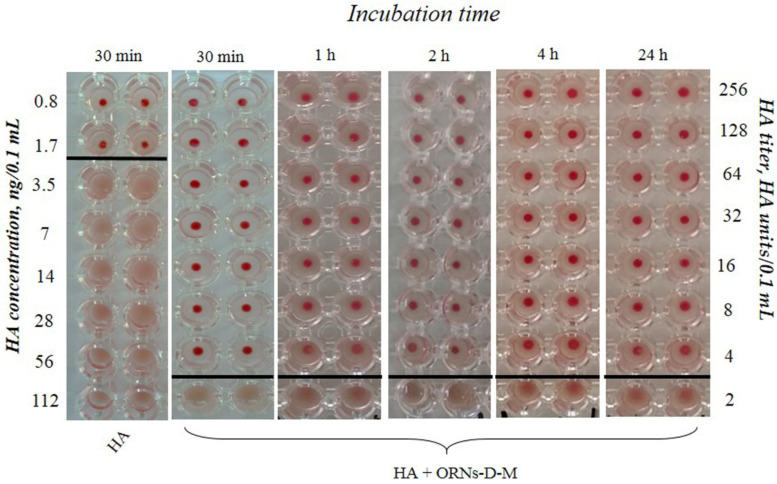
View of the hemagglutination reaction mixtures displayed in a round-bottomed microwell plate after the addition and dilution of the HA (Vaxigrip), which was pre-incubated with the ORNs-D-M (3.5 mg/mL) at the different duration of the experiment in hours. The HA titer was tested by HA assay. The picture shows one of five independent experiments.

**Figure 4 membranes-11-00757-f004:**
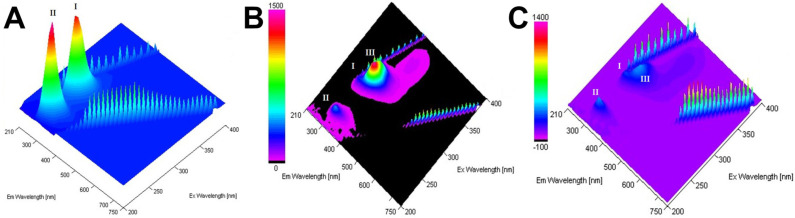
Total fluorescence spectra of the HA (**A**), HA–ORNs (**B**), and HA–ORNs-D-M (**C**) represented in the three-dimensional coordinate system. The amplitude of peaks presented according to the scale panel shown on the left side of the spectrum.

**Figure 5 membranes-11-00757-f005:**
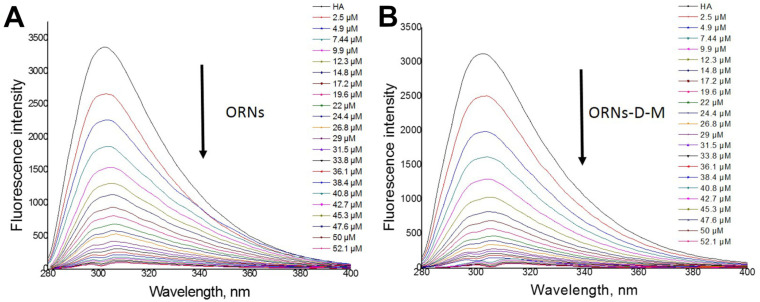
Quenching fluorescence of the hemagglutinin under titration with ORNs (**A**) and ORNs-D-M (**B**) ligands.

**Figure 6 membranes-11-00757-f006:**
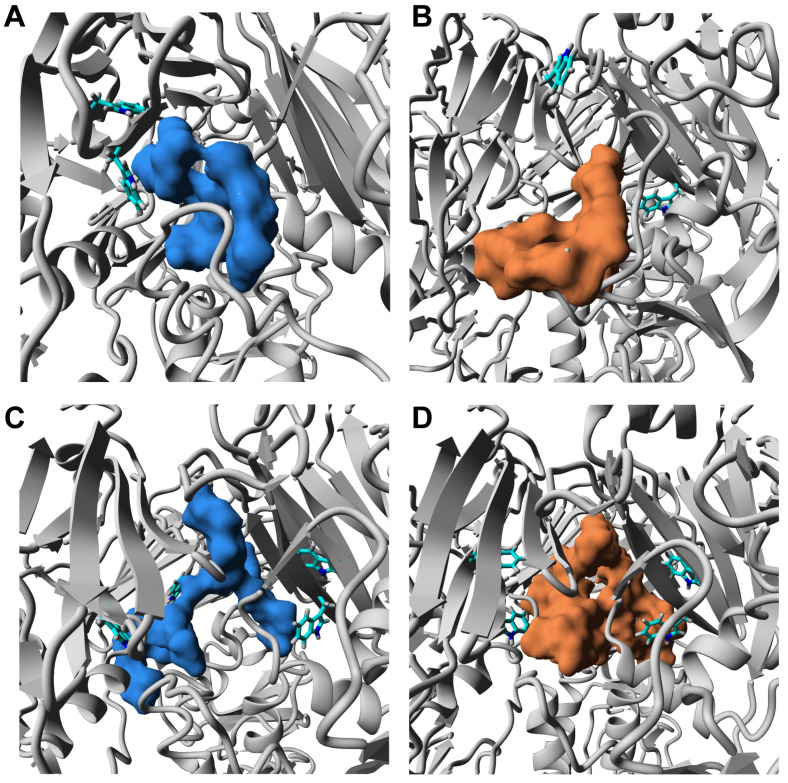
Close view position of the ORNs and ORNs-D-M ligands which non-specifically bind to the RBD domain. The HA3 complex with A3-ORNs (**A**), A3-ORNs-D-M (**B**), A6-ORNs (**C**), and A6-ORNs-D-M (**D**) ligands. The ORNs and functionalized ORNs-D-M ligands are highlighted as blue and brown, respectively. The side-chain of tryptophans (Trp180, Trp234) in HA_1_ subunits, detected as a part of the binding interface, are shown as a stick.

**Figure 7 membranes-11-00757-f007:**
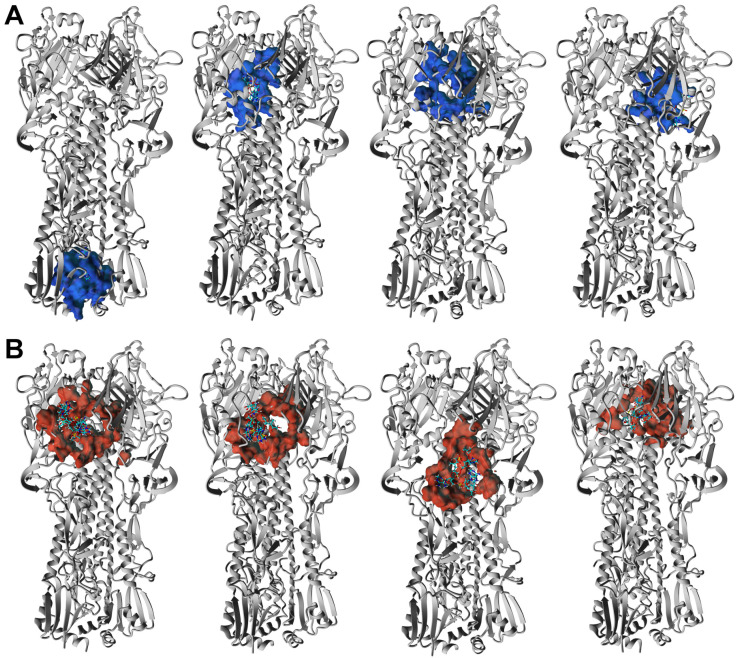
3D structure of the four lowest energy complexes of the HA3 with A3-ORNs (**A**) and A3-ORNs-D-M (**B**) ligands. The surface of the HA3 residues involved in interactions with A3-ORNs and A3-ORNs-D-M substrates is highlighted as blue and red, respectively.

**Figure 8 membranes-11-00757-f008:**
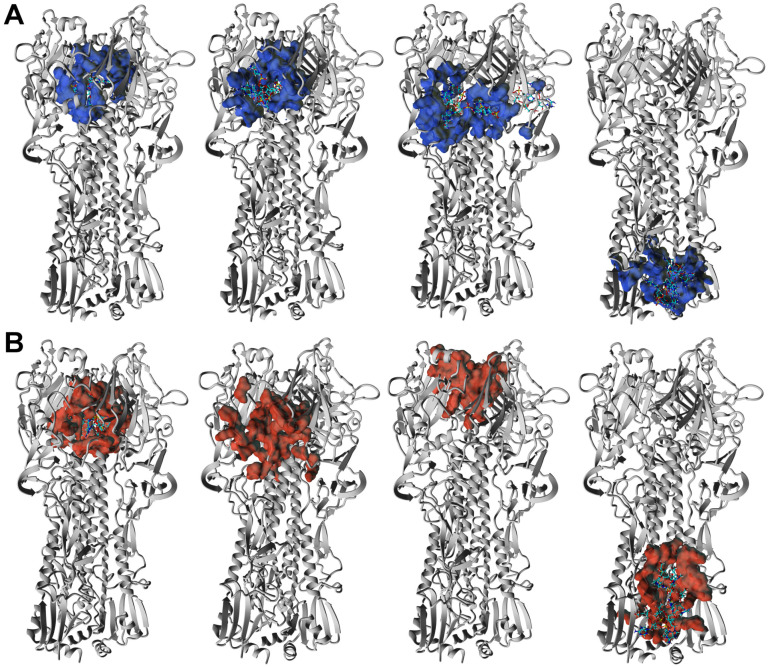
3D structure of the four lowest energy complexes of the HA3 with A6-ORNs (**A**) and A6-ORNs-D-M (**B**) ligands. The hemagglutinin epitops involved in interactions with A6-ORNs and A6-ORNs-D-M are highlighted as blue and red, respectively.

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
