# Peer review of "The Natural Oligoribonucleotides Functionalized by D-Mannitol Affected Interactions of Hemagglutinin with Glycan Receptor Indicating Anti-Influenza Activity"

_membranes, 2021, doi:10.3390/membranes11100757_

Round 1
Reviewer 1 Report
- The authors did experiment to prove the binding of ORNs and ORNs-D-M to influenza HA protein, and ORNs and ORNs-D-M decreases influenza HA titer. However, these experiments didn't support that ORNs and ORNs-D-M have antiviral activity against influenza. It's better to directly test ORNs and ORNs-D-M effect on influenza viral replication and quantify infectious viral amount, a plaque assay in cell culture is highly recommended.
- Language needs to be improved. Such as line46-47, line 51, etc.
Author Response
Point 1: The authors did experiment to prove the binding of ORNs and ORNs-D-M to influenza HA protein, and ORNs and ORNs-D-M decreases influenza HA titer. However, these experiments didn't support that ORNs and ORNs-D-M have antiviral activity against influenza. It's better to directly test ORNs and ORNs-D-M effect on influenza viral replication and quantify infectious viral amount, a plaque assay in cell culture is highly recommended.
Response 1: We already studied antiviral activity of the ORNs and functionalized by D-mannitol ORNs-D-M against influenza on cell culture. These data have been published in Melnichuk et al. Pharmaceuticals, 2017) cited as position 16 in the list of references. According to the suggestion of another reviewer, we added an article published before this one and described our initial experiments (Tkachuk et al. Rep. Natl. Acad. Sci. Ukr., 2010) placed in the list of references on position 17. The information was added to the Introduction.
Point 2: Language needs to be improved. Such as line46-47, line 51, etc.
Response 2: We provide the language corrections noted by the reviewer. We also perform additional work to improve the language of the manuscript.
Reviewer 2 Report
This work is in follow up to their previous papers (Pharmaceuticals 2017, 10, 71; Dopovidi Natsional’noi Akademii Nauk Ukraini 2018, (1), 92-99) where they showed the effect of oligoribonucleotides with D-mannitol against hemagglutinin of influenza virus. Although authors added more data such as fluorescence spectroscopy and molecular docking simulations in this manuscript, this point diminish the novelty and significance of present study. Also this manuscript had several lacks and mistakes as following:
- In the title and text, authors used the word “human hemagglutinin”. I think it’s better to use just hemagglutinin because of its origin.
- Authors have to add a reference for their previous study (Dopovidi Natsional’noi Akademii Nauk Ukraini 2018, (1), 92-99).
- In the text, there are “statistical analysis” in the Materials and Methods. Where is data under statistical analysis. Authors should add data with statistical analysis.
- In the line 42, page 2, please change the order as “viral binding, entry, and fusion”.
- In the line 17, page 1, please make correction aniti-influenza to anti-influenza.
- In the line 119, page 5, please add “,” between constant (Kd) and we.
- Authors wrote Kd values for both substrates as 10-7 – 10-6 M and “similar results were obtained with the molecular docking procedure” in the lines 186 – 189. However, the docking procedure gave different Kd values like 10-6 – 10-4 M (in the lines 209 – 213). Please check data and make correction.
- In the legend of Figure 1, authors should remove one of HA or hemagglutinin in the first line.
- The legend of Table S1: ORN and ORN-D-M --> ORNs and ORNs-D-M
- The legend of Figure S1: (B) A3-ORNS-D-M --> (B) A3-ORNs-D-M
- Authors should unify –DM to –D-M in the legend of figure S2.
- In the legend of figure S2, please revise “ complexes ()HA-ONs and ORNs-DM)” to “ complexes (HA-ORNs and ORNs-D-M)”
- In the figure S3, please use just one unit for Kd±SD as 12.1 ± 0.46 μM and 9.54 ± 0.58 μM instead of 12,1 μM ± 0,46 μM and 9,54 μM ± 0,58 μM.
- Please check the sentence of lines 155 – 156.
- In the line 157, page 5, ~56 ng/ml --> ~56 ng/0.1 ml
- From the figure S4, authors performed the docking simulations for complexes of the HA3 hemagglutinin with 32 substrates of A6-ORNs-D-M. Authors should provide the information for 32 substrates.
- Authors should explain in detail why they chose A3-ORNs, A3-ORNs-D-M, A6-ORNs and A6-ORNs-D-M ligands for docking.
Author Response
Point 1: In the title and text, authors used the word “human hemagglutinin”. I think it’s better to use just hemagglutinin because of its origin.
Response 1: Thank you for the suggestion. We agree that hemagglutinin has to use due to the origin of the protein.
Point 2: Authors have to add a reference for their previous study (Dopovidi Natsional’noi Akademii Nauk Ukraini 2018, (1), 92-99)
Response 2: We add the reference to the list of references under position 17 and cited this article in the Introduction.
Point 3: In the text, there are “statistical analysis” in the Materials and Methods. Where is data under statistical analysis. Authors should add data with statistical analysis.
Response 3: Since we used the HA protein with the same concentration, using Student's t-test is inappropriate. So we decided to remove this section from the Materials and Methods.
Point 4: In the line 42, page 2, please change the order as “viral binding, entry, and fusion”
Response 4: corrected
Point 5: In the line 17, page 1, please make correction aniti-influenza to anti-influenza
Response 5: corrected
Point 6: In the line 119, page 5, please add “,” between constant (Kd) and we.
Response 6: corrected
Point 7: Authors wrote Kd values for both substrates as 10-7 – 10-6 M and “similar results were obtained with the molecular docking procedure” in the lines 186 – 189. However, the docking procedure gave different Kd values like 10-6 – 10-4 M (in the lines 209 – 213). Please check data and make correction.
Response 7: Thank you for this note. We do not expect that results of computer simulations will be possible to direct comparison with experimental data. In our opinion, it is interesting to note that the docking procedure demonstrates a moderate level of dissociation constant. Nevertheless, taking into account that that point is discussed shortly in subsection 3.3 ‘ORNs and ORNs-D-M bind to the HA protein mostly in RBD subdomain’ in the following form (lines 209 - 213): “The Kd value obtained in the docking procedure is in the range of 10(-8) - 10(-4) M suggested that interactions ORNs and ORNs-D-M with hemagglutinin is not specific and characterized by the moderate level of affinity (Table S1 and S2). Extracted Kd values are in line with previously on base experimental data recorded by fluorescence spectroscopy (Figure S3).” Taking in mind discussion in subsection 3.3 we remove the sentence “similar results were obtained with the molecular docking procedure” from subsection 3.2 ‘HA fluorescence quenching by the ORNs and ORNs-D-M’ (lines 186 – 189 submitted manuscript).
Point 8: In the legend of Figure 1, authors should remove one of HA or hemagglutinin in the first line.
Response 8: corrected
Point 9: The legend of Table S1: ORN and ORN-D-M --> ORNs and ORNs-D-M
Response 9: corrected
Point 10: The legend of Figure S1: (B) A3-ORNS-D-M --> (B) A3-ORNs-D-M
Response 10: Corrected
Point 11: Authors should unify –DM to –D-M in the legend of figure S2.
Response 11: Corrected
Point 12: In the legend of figure S2, please revise “ complexes ()HA-ONs and ORNs-DM)” to “ complexes (HA-ORNs and ORNs-D-M)”
Response 12: Corrected
Point 13: In the figure S3, please use just one unit for Kd±SD as 12.1 ± 0.46 μM and 9.54 ± 0.58 μM instead of 12,1 μM ± 0,46 μM and 9,54 μM ± 0,58 μM.
Response 13: Corrected version of Figure S3 provided
Point 14: Please check the sentence of lines 155 – 156.
Response 14: The sentence on lines 155 – 156 is corrected, the correct concentration of HA in our experiments was 3.5 ng/ml.
Point 15: In the line 157, page 5, ~56 ng/ml --> ~56 ng/0.1 ml
Response 15: Leave as a submitted previously manuscript. The correct HA concentration is ~56 ng/ml.
Point 16: From the figure S4, authors performed the docking simulations for complexes of the HA3 hemagglutinin with 32 substrates of A6-ORNs-D-M. Authors should provide the information for 32 substrates.
Response 16: Our AutoDock Vina procedure included 5 runs docking simulations with 32 substrates each. The text subsection 2.4 ‘Docking of the ORNs and ORNs-D-M to influenza hemagglutinin’ in Materials and Methods was corrected. The evaluated data obtained in one run on simulations for the HA3 hemagglutinin with 32 substrates are provided in Supplementary Materials. Due to big amount of data we prepare two tables – Table S1 for A3-ORNs / A3-ORNs-D-M and Table S2 for A6-ORNs / A6-ORNs-D-M.
Point 17: Authors should explain in detail why they chose A3-ORNs, A3-ORNs-D-M, A6-ORNs and A6-ORNs-D-M ligands for docking.
Point 17: The ORNs and ORNs-D-M with three and six nucleotides were selected based on our initial data according to the distribution of length in the mixture. Also, our computational possibility required quite a lot of time for the docking ORNs and ORNs-D-M longer than six nucleotides. In such a case, the selection of ORNs and ORNs-D-M comprised of three and six nucleotides seems to be a reasonable choice to extract structural information about HA complexes with different lengths of substrates. We correct the text in subsection 3.3 ‘ORNs and ORNs-D-M bind to the HA protein mostly in RBD subdomain’ in the following redaction (lines 188 - 198):
‘According to our preliminary data, the distribution of oligoribonucleotides reveals variation from one up to 20 nucleotides or even longer. Nevertheless, the majority of the ORNs comprise between three and nine units with maximum populated chains with six nucleotides. Length of the ORNs and functionalized ORNs-D-M can be an additional parameter regulated process of binding substrates to the HA protein, which we explored by the docking simulations with the AutoDock Vina algorithm [18]. Taking into account distribution of chains length there are two types of ligands, with three (A3-ORNs) and six (A6-ORNs) nucleotides, together with their functionalized by D-mannitol (A3-ORNs-D-M, and A6-ORNs-D-M) variants were selected as a reasonable choice for representation of whole ORNs and ORNs-D-M ensemble (Figure S1).’
Round 2
Reviewer 2 Report
I read carefully this revised manuscript entitled “The natural oligoribonucleotides functionalized by D-mannitol affected interactions of hemagglutinin with glycan receptor indicating anti-influenza activity”. Authors responded and corrected for most of the comments I had pointed out. Thus, I recommend this work to be published in the “membranes” after minor alterations as follows:
For the comment 15, authors responded as “Leave as a submitted previously manuscript. The correct HA concentration is ~56 ng/mL”. However, authors have to check the unit at left side in the figure 2. There is a unit as ng/0.1 mL. Which one is right? Also, check the unit in the figure 3.
In the line 153, page 5, please change “1.7–3.5 mg/ml” to “1.7 and 3.5 mg/ml” because authors used just two concentrations. This is not suitable for use as a range.
Author Response
Point 1. For the comment 15, authors responded as “Leave as a submitted previously manuscript. The correct HA concentration is ~56 ng/mL”. However, authors have to check the unit at left side in the figure 2. There is a unit as ng/0.1 mL. Which one is right? Also, check the unit in the figure 3.
Response 1: Thanks for your point. We double-check the concentration of the HA in our experiments through the whole manuscript. The following corrections were provided:
line 81 – The HA (225 ng/0.1 mL) was incubated
line 93 – the HA (225 ng/0.1mL) was incubated with
Line 142 – isolated HA at a concentration of 225 ng/0.1 ml was 65 HAU/0.1 ml (Figure 2)
Line 151 – we estimated the minimum concentration of HA – 3.5 ng/0.1 ml …
Line 153 – concentration of 1.7–3.5 mg/ml interact with ∼56 ng/0.1ml ...
Point 2. In the line 153, page 5, please change “1.7–3.5 mg/ml” to “1.7 and 3.5 mg/ml” because authors used just two concentrations. This is not suitable for use as a range.
Response 2: corrected